# Phosphoramidate Azole Oligonucleotides for Single Nucleotide Polymorphism Detection by PCR

**DOI:** 10.3390/ijms25010617

**Published:** 2024-01-03

**Authors:** Alexey S. Chubarov, Elizaveta E. Baranovskaya, Igor P. Oscorbin, Ivan I. Yushin, Maxim L. Filipenko, Dmitrii V. Pyshnyi, Svetlana V. Vasilyeva, Alexander A. Lomzov

**Affiliations:** Institute of Chemical Biology and Fundamental Medicine, SB RAS, 8 Lavrentiev Avenue, 630090 Novosibirsk, Russia; evgenevnalizaveta@gmail.com (E.E.B.); osc.igor@gmail.com (I.P.O.); i.yushin@g.nsu.ru (I.I.Y.); max@niboch.nsc.ru (M.L.F.); pyshnyi@niboch.nsc.ru (D.V.P.); svetlana2001@gmail.com (S.V.V.)

**Keywords:** PCR, allele-specific PCR, modified oligonucleotides, phosphoramide benzoazole oligonucleotides, PABAO, mutation detection, single nucleotide polymorphism detection, SNP, *KRAS* mutations

## Abstract

Detection of the Kirsten rat sarcoma gene (*KRAS*) mutational status is an important factor for the treatment of various malignancies. The most common *KRAS*-activating mutations are caused by single-nucleotide mutations, which are usually determined by using PCR, using allele-specific DNA primers. Oligonucleotide primers with uncharged or partially charged internucleotide phosphate modification have proved their ability to increase the sensitivity and specificity of various single nucleotide mutation detection. To enhance the specificity of single nucleotide mutation detection, the novel oligonucleotides with four types of uncharged and partially charged internucleotide phosphates modification, phosphoramide benzoazole (PABA) oligonucleotides (PABAO), was used to prove the concept on the *KRAS* mutation model. The molecular effects of different types of site-specific PABA modification in a primer or a template on a synthesis of full-length elongation product and PCR efficiency were evaluated. The allele-specific PCR (AS-PCR) on plasmid templates showed a significant increase in analysis specificity without changes in Cq values compared with unmodified primer. PABA modification is a universal mismatch-like disturbance, which can be used for single nucleotide polymorphism discrimination for various applications. The molecular insights of the PABA site-specific modification in a primer and a template affect PCR, structural features of four types of PABAO in connection with AS-PCR results, and improvements of AS-PCR specificity support the further design of novel PCR platforms for various biological targets testing.

## 1. Introduction

One of the most frequent DNA variations is single-nucleotide polymorphisms (SNP), where one nucleotide is replaced by another. SNP can cause various diseases, such as cancer, diabetes, heart disease, etc. In this regard, SNPs are very effective disease markers. Single-nucleotide exchanges can not only be germinal, e.g., SNP, but also occur as mutations in somatic cells which is an important factor of cell malignization. However, it is difficult to identify the smallest genetic difference in a low amount of DNA with a high specificity, which may lead to incorrect diagnosis and delayed treatment. The Kirsten rat sarcoma gene (*KRAS*) is a crucial gene participating in the tumorogenesis of various cancers, such as pancreatic, colon, lung, and ovarian cancer [1,2,3,4]. As the most frequently mutated oncogene, *KRAS* has attracted numerous studies on its initiation, progression, and challenging treatment [4]. According to the COSMIC database (the Catalogue Of Somatic Mutations In Cancer), the most common *KRAS*-activating mutations are located in codons 12–13, as p.G12V, p.G12A, p.G12D, etc. Detection of *KRAS* mutations and mutation allele frequency (MAF) is essential in terms of targeted anti-cancer therapy [4]. A variety of mutation detection methods are available for *KRAS* mutation detection, including allele-specific (AS) real-time polymerase chain reaction (AS-PCR), droplet digital PCR, melting curve analysis, next-generation sequencing (NGS), etc. [1,2,5]. Among these techniques, NGS technology provides simultaneous detection of multiple mutations with high sensitivity and specificity [6,7,8]. However, NGS is a time-consuming, expensive, and laborious method, which requires highly qualified personnel. In clinical practice, rapid, cheap, simple, and routine methods are preferred for large numbers of typical tests. In principle, PCR assays have firmly occupied this area of analysis, providing reliable results with limited resources.

AS-PCR utilizes specific oligonucleotide primers to amplify target mutated DNA on the background of wild-type (WT) DNA. In this way, crucial for the overall success of an AS-PCR experiment is the careful design of synthetic oligonucleotide primers. However, the AS primer choice is challenging in order to achieve high sensitivity, selectivity, and specificity. AS primers contain 1–2 additional mismatches at the 3′-terminus of the primer to increase the specificity of SNP detection [9,10,11,12]. Typically, the design takes several testing iterations with numerous possible primers to optimize assay performance. In many cases, it is not enough to achieve good discrimination between mutant and wild-type DNA [2,9]. In this way, crucial for the overall success of an AS-PCR experiment is the careful design of synthetic oligonucleotide primers. Over the previous decades, primers with a chemical modification or nucleic acid analogs have been used extensively to enhance the specificity of AS-PCR analysis [2,13,14,15,16,17,18,19].

In recent years in our laboratory, a new type of uncharged phosphate-modified nucleic acid analog, namely phosphoryl guanidine oligonucleotides (PGOs), has been described [20,21]. The uncharged-phosphate PG modification can easily be introduced into any position of an oligonucleotide via standard automatized phosphoramidite oligonucleotide synthesis. PG-modification protects therapeutic siRNA from degradation [22,23,24,25,26] and increases the sensitivity and specificity of *KRAS* and PIK3CA mutation detection by AS-PCR [21,27]. The latter showed 100% specificity on formalin-fixed paraffin-embedded (FFPE) tissues. However, the sensitivity was a bit lower, 78–86%, due to the presence of several false-negative results (for the samples with MAF 0.05–0.56%). It may be explained by a lower PCR efficiency for PG-modified primers with 3′-terminus mismatches [27]. In this connection, in our institution, analogs of PG, the internucleotide benzoazole phosphoramide (PABA) substitutions, were developed [28]. PABA groups are bulkier and more hydrophobic than PG. N-benzimidazole and 1,3-dimethyl-N-benzimidazole have uncharged internucleotide phosphates under physiological conditions, and N-benzoxazole and N-benzothiazole are partially charged [28]. The possibility of automatic synthesis of novel PABA oligonucleotides on an NA synthesizer and a variety of physicochemical properties without significant changes in secondary structure [28,29] explains their choice for this study. It can be proposed that the use of PABAO makes it possible to intentionally vary biological properties for different applications.

Here, for proof-of-concept studies, we have chosen four modifications: internucleotide N-benzimidazole, N-benzoxazole, N-benzothiazole, and 1,3-dimethyl-N-benzimidazole (benzoazoles) phosphoramide (PABA) groups (Figure 1). We introduced PABA modification into different sites of the model oligonucleotide served as a primer or a template and performed a PCR reaction. Molecular effects of site-specific PABA modification on the synthesis of a full-length elongation product and PCR efficiency were studied. We used phosphoramide benzoazole oligonucleotides (PABAOs) as primers for AS-PCR and assessed the specificity of AS-PCR on a model of the most common *KRAS* mutations G12A and G12V, using plasmid templates. In summary, PABA-modified primers increased the specificity of PCR without an efficiency loss, which shows the great potential of the new oligonucleotides with PN linkage.

## 2. Results and Discussion

### 2.1. Effect of PABA Modification Position at the 3′-Terminus of the Primer on the Elongation Using Taq DNA Polymerase

The influence of the number of PABA groups and their position at the 3′-terminus were investigated on a template/primer model (see Section 3.2, Table 1). The primer sequence of 20 bases was chosen as 5′-FAM-GGTGCGCTCCTGGACGTAGC-3′ (marked as Z-oligonucleotide). The FAM dye was inserted for fluorescence detection of primer and reaction product using gel electrophoresis assay. Several Z oligonucleotides with one PABA modification in 1–7 phosphates, and two modifications in 1,2 and 1,3 phosphates were synthesized (Table 1). A template of 30 bases was used with a sequence of 5′-CTGTTGTTTAGCTACGTCCAGGAGCGCACC-3′ (marked as T-oligonucleotide). Several template sequence variations were synthesized (Table 1). Bold symbols marked nucleotides represent mismatched nucleotides in relation to the Z sequence.

Before elongation using Taq DNA polymerase, oligonucleotide complexes between Z-primer and T-template were analyzed using thermal denaturation analysis, known as melting curve analysis, which assesses the thermodynamic stability of complexes. Typical melting data are presented in Appendix A. To characterize the qualitative helix-to-coil transition of PABAO/DNA and DNA/DNA duplexes, melting temperature (Tm) was used. The Tm value is close to the maximum of differential melting curves and could be easily determined [30]. A comparison of the Z/T series thermal stability indicates a slight difference in native and modified complexes, both for the full match and containing mismatches. The Tm values of the Z/T series are in the range of 72.5 ± 0.3 °C (for Z1,3-O/T1,2) to 77.6 ± 0.8 °C (for Z1-D/T). However, the average Tm value for the selection was 74.5 °C with a range of values of 1.1 °C (Appendix A). Analysis of thermodynamic parameters of the Z/T-series indicates similar values of Gibbs free energy changes (∆G°_37_). The ∆G°_37_ ranges from −22.4 ± 1.5 kcal/mol (Z1,3-D/T1) to −24.6 ± 2.5 kcal/mol (Z5-S/T), which is close to the experimental error (Appendix A). For primer with two PABA modifications, Z1,2-X/T and Z1,3-X/T complexes (X = N, S, O, and D), slightly lower ∆G°_37_ values were evaluated, which indicates negligible decreases in the stability of the complexes. Furthermore, no significant destabilization was observed for duplexes with two PABA modifications of any type and terminal mismatches (Z1,2-X/T1,2 and Z1,3-X/T1,2). The same observation was found for enthalpy (∆H°) changes (Appendix A). The ∆H° changed from 105 ± 5 kcal/mol (for Z1,3-D/T) to −154 ± 22 kcal/mol (for Z5-S/T), which differed by 20–30% and was higher than the experimental error values. Nonetheless, the enthalpy–entropy compensation [31] resulted in a significantly lower variation in integral thermodynamic characteristics ∆G°_37_ and Tm.

The elongation with the Z/T series was carried out using Taq polymerase and the following procedure: 1 cycle of heating at 95 °C for 2 min, primer annealing at 30 °C for 1 min, and extension at 60 °C for 7 min. The reaction was stopped by 2% LiClO_4_ in acetone with subsequent analysis of reaction mixtures by gel electrophoresis. The results are presented in Figure 2, Figure 3, Figure 4 and Figure 5. The charts in Figure 2, Figure 3, Figure 4 and Figure 5 represent data for four PABA modifications. The examples of gel electrophoresis images stained by StainsAll and fluorescence detection are presented for S-modification (on the right in Figure 2, Figure 3, Figure 4 and Figure 5). The data for O-, N-, and D-PABAO can be found in Appendix A (Appendix A).

The elongation data for fully complementary complexes Z/T show efficient elongation, almost 100% in most cases, for all PABA modifications in phosphates from 1 to 7 in Z primer (Figure 2). For primers with two modifications, Z1,2 and Z1,3, slight inhibition was observed with N- and D-PABA. However, the elongation product yield becomes much lower in the system Z/T-1 with a 3′-end mismatch (Figure 3). The yield for an unmodified primer Z0 is ~63%. For the modified primers, the same amount of product was detected only with Z7-O, S, and N /T-1 mixtures. For Z7-D/T-1, the yield decreases to 40%. It seems that the 7-phosphate’s modification has minimal effects. The same data were obtained for the Z/T2 duplexes (mismatch in the second base) and Z/T1,2 duplexes (two mismatches) (Figure 4 and Figure 5). For Z/T1 (Figure 3), the most efficient elongation was shown for PABAOs with O- and S-modification. Furthermore, among 1–6 phosphates, the third position provided sufficient elongation product for several modifications. Previously, it was shown that methyl phosphotriester oligonucleotide modification in 3- and 7-positions does not affect DNA-Taq polymerase interactions and protein motions during the PCR [32]. For SNP detection, the 3′-end of the AS primer is fully complementary to the mutated DNA (Z/T system) and has a 3′-end mismatch with WT DNA (Z/T-1). In the presence of a mutation, the reaction in Z/T should occur much more efficiently than in the Z/T1 complex. It may be seen that without a modification, the reaction proceeds efficiently in both cases (Figure 2 and Figure 3). However, the presence of the modification blocks Z/T1 complex elongation and does not influence Z/T, which is excellent for SNP detection.

Concerning the systems Z/T2 and Z/T1,2, a low elongation efficiency, less than 5%, occurred for one cycle of reaction (Figure 4 and Figure 5). These models simulate a situation with a mismatch in a second base and two mismatches. Generally, a single 3′-end mismatch is not enough for good discrimination between WT and mutated DNA. The presented results in Figure 4 and Figure 5 clearly show an increase in specificity in modified primers towards the native primer (control column in chart). The reaction is inhibited with a non-matched template in both cases.

Finally, these results show that a PABA modification in 1–7 phosphate from primer’s 3′-end of primer does not influence the elongation reaction for a fully complementary primer/template complex (Figure 2). For the partly mismatched 3′-terminus primer/template complex, PABA modification in 1–6 phosphate from primer’s 3′-end of primer decreases elongation efficiency more than the unmodified primer (Figure 3, Figure 4 and Figure 5). However, N- and D-modifications almost stop PCR, which may be a problem, and leads to PCR inhibition. This requires additional extensive research on other AS-PCR models. For clinical samples, such strong inhibition could be detrimental only if a low amount of mutated DNA is in the background of a high excess of WT DNA.

### 2.2. Effect of PABA Modification Position in the Template on the Elongation

Despite numerous studies on the effect of primer modification [2,14,21,27,32,33], limited studies have been carried out on the effect of template modification. However, this can be very important for many modern PCR applications or complex amplification approaches such as LAMP [34,35]. For example, for SNP, detection primer with chemical modification may be used (e.g., Section 2.3). In a PCR reaction, modified primer binds with one DNA chain, and the product is synthesized by primer elongation. In this way, the new DNA chain contains chemical modification, which can affect further PCR process, e.g., inhibit it. Moreover, it may lead to specificity and selectivity of analysis changes. Therefore, the effect of chemical modification in the template may show molecular insights into the elongation process, which allows us to study the mechanism.

The influence of the number of PABA groups and their position in the template were investigated on a model template/primer. For these studies, a Z oligonucleotide of 20 bases served as a template, and a P-oligonucleotide of 8 bases (5′-Cy5-GCTACGTC-3′) was used as a primer (see Section 3.2, Table 1). The Cy5 dye was inserted for fluorescence detection of primer and reaction product using gel electrophoresis assay.

As well as Z/T duplexes, Z/P-series were analyzed using the UV-meting approach. The Z/P series exhibits different stability for one or two PABA modifications in Z oligonucleotides (Appendix A). However, it is impossible to determine thermodynamic parameters due to the smoothly increasing melting curves (Appendix A). The estimated Tm values determined as a maximum of the differential melting curve were in the range of 40–45 °C for either unmodified or a single modification, whereas the thermal stability of double-modified duplexes was reduced to 35–40 °C. This decrease in Tm values agrees with our previous results on the thermal stability of PABAO [28,29,32].

Elongation of the Z/P-series was carried out using Taq polymerase and the following procedure: three cycles of heating at 95 °C for 2 min, primer annealing at 30 °C for 1 min, and extension at 37 °C for 7 min. Three cycles instead of one for the Z/T system were used due to the low reaction efficiency. The reaction was stopped by 2% LiClO_4_ in acetone with subsequent analysis of reaction mixtures by gel electrophoresis. The results are presented in Figure 6. The chart in Figure 6 represents data for four PABA modifications. The examples of gel electrophoresis images stained by StainsAll and fluorescence detection are presented for S-modification (on the right in Figure 6). The data for O-, N-, and D-PABAO can be found in Appendix A.

The PABA modification in the Z template significantly influences PCR since they are in the DNA/Taq polymerase contact zone [32]. The modification might affect DNA/Taq complex conformation and stability, the motion of the enzyme, and the ability to shift the enzyme along the chain of DNA [32]. The most efficient sites were 3 and 7 due to the phosphates’ direction away from protein and high template chain motion possibilities [32]. On the contrary, PABA modification in positions 1, 2, and 4 provided less than 40% of elongation product yield. It may be explained by a disturbance in the protein structure through the interaction of polymerase with PABA [32]. The two PABA modifications do not lead to a high amount of elongation product due to the possible decrease in DNA duplex thermal stability. Thereby, modifications in a template can highly influence PCR, which could be useful for novel PCR approaches.

### 2.3. Effect of PABA Groups Position at the 3′-Terminus of the Primer on the AS-PCR Results

Real-time AS-PCR has been widely used for SNP detection [36,37,38,39]. To obtain reliable discrimination between WT and mutated DNA, AS primers usually have a 3′-end and one additional mismatched base pair within a 2–4 nucleotide region on the 3′-terminus [9,10,40]. To examine the usefulness of PABAO as primers for AS-PCR, we utilized previously established primers’ sequences for *KRAS* mutation detection [21]. The full sequence of primers and qPCR conditions are presented in Section 3.4. The primers are coded as four nucleotides of the primer 3′-terminus with the “*” symbol as a position of PABA modification. We have used primers CTG**T** and CTG**C** with 3′-end mismatches to WT DNA, and fully matched to mutated DNA. The second primers’ type (CT**TC** and CT**AT**) has two mismatches to WT DNA and one mismatch to mutated DNA. Plasmids harboring *KRAS* gene fragments of WT and sequences with G12A and G12V mutations served as templates (see Section 3.3). For these mutations, SNP is G/C and G/T, respectively. Primers for the G12A mutation demonstrated much higher discrimination efficiency between WT and mutated DNA due to the difference in G12V 3′-end mismatch concerning WT DNA [21]. However, increasing the specificity of “unfavorable” mismatch detection, such as G12V *KRAS*, is highly important for clinical diagnostics. To test PABA modification on AS-qPCR, we have introduced four types of PABA modification in 2–4 internucleotide phosphates in AS primers for G12A and G12V *KRAS* mutation detection (Table 2). Typical amplification curves obtained from AS-PCR of the unmodified and modified primers are presented in Figure 7.

Commonly, for G12A, the CT**TC** primer with all modifications provided Cq of N/A for WT DNA samples and high discrimination efficiency (see ΔCq column, Table 2). However, for 1% mutated DNA, the Cq values of N- and D-modified primers increase by 2–3 cycles compared to the unmodified primer. For O- and S-PABA modifications from 2 to 4 phosphate the PCR efficiency was similar to the unmodified CT**TC** primer (Table 2). For PABA-modified CTG**C** (G12A), the ΔCq values are much lower and, in most cases, similar to the native CT**TC** primer. It seems that PABA modifications serve as a mismatch-like disturbance and increase primer specificity. Moreover, modifications of the second or fourth phosphates higher increase discrimination efficiency than in the third phosphate. Together, these results provide important insights into possible simplified primer design with 3′-end mismatch and PABA modification instead of combinatory selecting of mismatch position and a nucleobase. Previously, similar results were obtained using a limited primer series for PG modification [21].

In contrast with CTG**C** (G12A), CTG**T** (G12V) showed noticeable discrimination efficiency with a modification in the fourth phosphate. For the CT**AT** primers, the results were similar to those for CT**TC** but less striking (Table 2). The third phosphate from the 3′-end is the optimal modification position, providing higher ΔCq and similar PCR efficacy with a 1% mutated DNA sample. However, for the second and fourth phosphates, the results are just a little worse. This finding is consistent with the optimal place for a PG modification in the third internucleotide phosphate [21].

For four types of PABA modifications in different AS primers, the PCR efficiency was calculated in PCR experiments with a serial dilution of the template (Table 3). PCR efficiency for the reference unmodified primer (k-ref) was 100%, indicating optimal reaction conditions. Primers with S- and O-modifications demonstrate PCR efficiency similar to the unmodified primer or higher. In many cases, for the N- and D-modifications, PCR efficiency is lower than 90%, which was previously observed for PG modification [21,27]. The low PCR efficiency makes the detection of low mutation percent impossible, which leads to false-negative results in clinical samples. It is not surprising that for these primers, high Cq values were obtained (Table 2). In this way, primers with D-modification and most of the N residues, such as CT***TC**, CT*G**C**, C*T**AT**, *CT**AT**, and CT*G**T,** should be excluded from further PCR experiments.

CT**TC** and CT**AT** primers were previously used for *KRAS* mutation detection [40,41], and that is why they were chosen for this study. However, unmodified primers do not provide high specificity, which may be estimated from ΔCq values of 5.3 and 3.1 for CT**TC** and CT**AT**, respectively (Table 2). For PG-modified primers C*T**TC** and C*T**AT,** the high ΔCq of 12.3 and 5.5 were obtained [21]. However, a PCR efficiency loss of 10% relative to CT**TC** primer was received for PG-C*T**TC** [21]. Herein, C*T**TC** and C*T**AT** with O- and S-PABA modification showed ΔCq of 12.6–12.7 and 5.3–6.6, respectively, without the PCR-efficiency loss, and good Cq values for 1% mutated DNA sample.

The qPCR data analysis is summarized in Table 4. Three parameters, PCR efficiency, ΔCq, and Cq values, were compared to the unmodified primer to give insights into the possible further use of PABAO for AS-PCR assay. We have chosen these parameters in order to detect mutations with enough specificity without the loss of sensitivity [21,27]. We divided the results into their categories and marked them as “not suitable values” (red color), “average” (yellow color), and “optimal or excellent” (green color). For example, the cutoff for PCR efficiency was chosen as lower 90% (red), 90–94% (yellow), and higher 95% (green). Amplification efficiency is an essential parameter. The high amplification efficiency allows for mutations in samples with a low MAF, which is common in clinical FFPE samples. Low PCR efficiency can lead to false-negative testing results. The ΔCq value is one of the most important parameters. High ΔCq means robust discrimination of WT and mutated DNA with a low probability of false-positive or false-negative test results. Finally, the Cq values should not be high, providing quick analysis and excluding false-positive results or inaccuracies. Of course, it will be excellent if primer modification would not enhance Cq values for 1% of mutated DNA samples but also provide high specificity. However, it is extremely difficult to obtain excellent results for all parameters. Previously, PG modifications increased PCR specificity for model DNA plasmid and clinical samples (FFPE tissues) [21,27]. However, it may be concluded that these primers were “too specific”, leading to PCR inhibition, high Cq values, and false-negative clinical samples [27]. In this way, the balance between these parameters is crucial for robust mutation detection. It may be better to use a primer with an average specificity and good PCR efficiency instead of a highly specific primer with a low PCR speed. According to the data in Table 4, O- and S-PABA modifications proved to be the most effective for further AS-PCR application. PABAO can enhance PCR specificity without loss of sensitivity by way of the synergetic effect with an additional mismatch, increasing the reliability of clinical AS-PCR assay. While preliminary, this discovery suggests that PABA modifications in some cases can be used with only 3′-end mismatches to increase primer specificity. The main principle of the developed methodology can be used to improve the PCR specificity regardless of the AS primer sequences. In summary, the assay is in the optimization stage and demonstrates the potential of PABAO oligonucleotides in the PCR analysis of biological samples. Further extended experiments with the detection of a low mutation amount, analysis of clinical samples, etc., are required to prove the initial concept experiment. There is still a lot of work left to be completed to confirm and reserve a place among the established oligonucleotide technologies.

To sum up, four types of PABAO were introduced into different sites of the model oligonucleotide served as a primer or a template, and then an elongation reaction was performed by Taq DNA polymerase. Single PABA modification in phosphates from 1 to 7 in primers for fully complementary primer/template complexes does not decrease elongation efficiency. For the primers with 3′-terminus mismatches, the influence of PABA on elongation is challenging (Figure 3, Figure 4 and Figure 5). However, the efficiency of elongation for PABAOs with O- and S-modification was much higher than for N and D. Among 1–6 phosphates, the third position provided sufficient elongation product for several modifications. It seems that 7-phosphate positions do not almost affect primer–Taq polymerase interactions. The PABA modification in the template significantly influences elongation (Figure 6), providing low elongation product yield in positions 1, 2, and 4 and high in 3 and 7. The two PABA modifications in the template lead to extremely low elongation product yield. Molecular effects of site-specific PABA modification in primer and template on a synthesis of a full-length elongation product provide valuable information for further PCR systems designed for various applications.

To show the proof of concept, we used PABAOs as primers for AS-PCR and assessed the specificity of AS-PCR on a model of the most common *KRAS* mutations p.G12A and p.G12V, using plasmid templates. Our results suggested that the PABA-modified primers with O- and S-modification have almost 100% PCR efficiency in most cases. N-PABA-modified primer has comparatively lower PCR efficiency, which highly depends on the modification site. D-PABA highly inhibited PCR. The incorporation of O- and S-PABA modification into PCR primers leads to the identification of p.G12A and p.G12V with good specificity (ΔCq value, Table 4) without a significant increase of Cq values of samples with mutated DNA (Table 4). The results on the model plasmid system provide a high potential of PABAOs for SNP detection with high specificity and selectivity, which further should be proved on clinical samples. In summary, PABA-modification potential may be used for establishing various PCR diagnostic systems.

## 3. Materials and Methods

### 3.1. Synthesis and Isolation of Oligonucleotides

Oligonucleotides were synthesized in an ASM-800 automated synthesizer (Biosset, Russia) according to the standard protocol of the 2-cyanoethyl phosphoramidite method using commercially available deoxyribonucleoside monomers and controlled porous glass (Glen Research, Sterling, VA, USA). Oligonucleotides carrying internucleotide N-(benzoazole)-phosphoramide moieties were synthesized using the protocol described previously [28]. The primary oxidation stage was replaced by the Staudinger reaction with N-benzimidazole, N-benzoxazole, N-benzothiazole, and 1,3-dimethyl-N-benzimidazole (benzoazoles) azides (0.25 M solution in acetonitrile). The purification of oligonucleotides by reverse phase high-pressure liquid chromatography (RP-HPLC) was performed on the Agilent 1200 series chromatograph (Agilent, Santa Clara, CA, USA) on a column (4.6 × 150 mm) containing the Eclipse XDB-C18 sorbent (5 μm) (Agilent, Santa Clara, CA, USA) with a 0–30% linear gradient of acetonitrile concentration in 0.02 M triethylammonium acetate solution for 40 min at a flow rate of 1.5 mL/min. Fractions containing the target product were evaporated in vacuo. The bulk of triethylammonium acetate was removed by coevaporations with ethanol. To remove the protecting dimethoxytrityl group, the purified oligonucleotides were treated with 80% acetic acid (25 °C, 7 min). Oligonucleotides were concentrated, followed by precipitation with 2% LiClO_4_ in acetone, washing with pure acetone, and desiccation under a vacuum. After desiccation, the oligonucleotides were dissolved in 0.05 mL of deionized water and stored at −20 °C. The concentration of oligonucleotides in solutions was determined on a UV-1800 spectrometer (Shimadzu, Kyoto, Japan) according to the procedure described earlier [28].

### 3.2. PCR

Reactions were performed in 20 µL containing 1× PCR-buffer (25 mM Tris-HCl (pH 8 at 25 °C), 50 mM KCl, 0.1 mM EDTA, 0.5% Tween-20, 0.5 mg/mL gelatin, 1 mM DTT, 6 mM MgCl_2_), 0.2 mM dNTP, 25 µM primers and matrix, and 2.5 U of Taq polymerase (Biosan, Novosibirsk, Russia). Elongation was carried out in the Eppendorf Mastercycle Classic (Eppendorf, Germany) according to the following program for the systems Z/T: 1 cycle of denaturation at 95 °C for 2 min, primer annealing at 30 °C for 1 min, and extension at 60 °C for 7 min (Table 1); elongation was also carried out according to the following program for the system Z/P: 3 cycles of denaturation at 95 °C for 2 min, primer annealing at 30 °C for 1 min, and extension at 37 °C for 7 min (Table 1). The reaction was stopped by 2% LiClO_4_ in acetone adding. Electrophoretic analysis of reaction mixtures was carried out in denaturing 15% polyacrylamide gel at pH 8.3. The gel was scanned on VersaDoc VH 4000 (Bio-Rad, Hercules, CA, USA) using the FAM or Cy-5 channel, and then visualization was performed by staining via StainsAll dye.

### 3.3. Plasmid Standards

The control plasmids contained a partial sequence of the wild-type *KRAS* gene or *KRAS* gene, with mutations in codon 12 (p.G12A, and p.G12V) serving as positive controls, and they were used to assess method sensitivity and were constructed by Shanghai RealGene Biotech, Inc. (Shanghai, China). Before use, all control plasmids were purified, linearized by digestion with BamHI restriction endonuclease, and quantified using a NanoDrop Lite A4 spectrophotometer (Thermo Fisher Scientific, Waltham, MA, USA).

### 3.4. Real-Time PCR

Real-time PCR assays were performed in 20 µL containing 1× PCR buffer (65 mM Tris–HCl (pH 8.9), 24 mM (NH_4_)_2_SO_4_, 0.05% Tween-20, 3 mM MgSO_4_), 0.2 mM dNTP, 300 nM primers, and 100 nM fluorescent hydrolysis probe, DNA template (exact amount indicated below), and 1 U of Taq polymerase (Biosan, Novosibirsk, Russia). A control plasmid was used as a DNA template at the concentration indicated below. Amplification was carried out in a CFX96 Real-Time PCR Detection System (Bio-Rad, Hercules, CA, USA) according to the following program: 95 °C for 3 min, followed by 45 cycles of 95 °C for 10 s, and 60 °C for 40 s with a collection of fluorescent signals at the FAM channel. Reactions were carried out at least in triplicate and performed several times on separate occasions.

Average Cq ± standard deviation (SD) values are given in the tables. PCR analysis was performed using a reverse primer (k-rev) 5′-CATATTCGTCCACAAAATGATTCTG-3′, probe 5′-FAM-CTGTATCGTCAAGGCACTCTTGC-BHQ1-3′ and a series of forward primers 5′-AAACTTGTGGTAGTTGGAGXXXX-3′. XXXX—four nucleotides of the primer 3′-terminus, which are presented in the text as an abbreviation of the whole primer. Boldly marked nucleotides represent mismatched nucleotides in relation to the wild-type DNA sequence. In each PCR plate, forward primer 5′-GACTGAATATAAACTTGTGGTAGTTG-3′ was used as a reference primer (k-ref) to compare the data between various PCR experiments. Corresponding ΔCq values were calculated and used for the further analysis of the primer’s efficacy. NTC—no template control.

### 3.5. UV Melting Experiments and Thermodynamic Analysis

Thermal denaturation assays of the oligonucleotide complexes were carried out in quartz cells (0.2 cm path length) by means of the Cary 3500 spectrophotometer (Agilent, USA) equipped multizone thermostabilized multicell holder. Melting curves were recorded at wavelengths 260, 270, and 330 nm in the 5−95 °C range at a temperature change rate of 0.5 °C/min. Absorbance at 330 nm was subtracted from the values at other wavelengths at each temperature (baseline). The maximum of the melting curve derivative was calculated as a melting temperature (T_m_). A 5 µM equimolar mixture of oligonucleotide samples was dissolved in the buffer: 10 mM sodium cacodylate (pH 7.2), 75 mM NaCl, and 6 mM MgCl_2_. Thermodynamic parameters of duplex formation (changes in enthalpy ΔH°, entropy ΔS°, and ΔG° at 37 °C) were determined by nonlinear fitting of denaturation and renaturation curves via the two-state model at 260 and 270 nm and averaged.

## Figures and Tables

**Figure 1 ijms-25-00617-f001:**
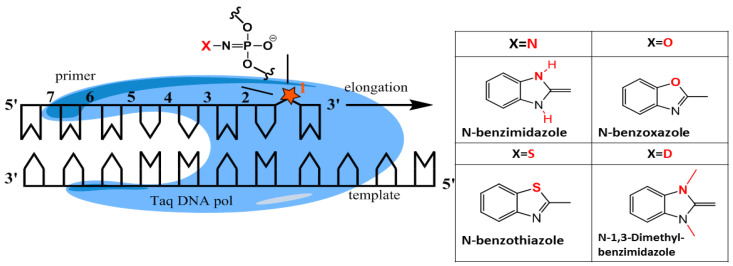
Taq DNA polymerase and template/primer complex schematic representation. The red star indicates phosphoramide benzoazole (PABA) modification of internucleotide phosphate moiety.

**Figure 2 ijms-25-00617-f002:**
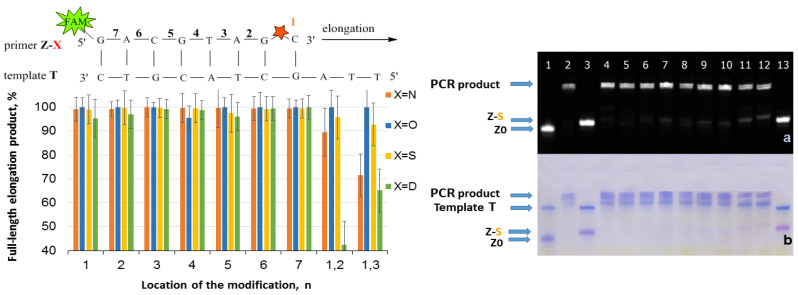
Effect of PABA modification location in a primer on elongation efficiency. (**Left**): Schematic representation of the Z-X/T fragment with a red star of PABA modification with number (n) of phosphates from 3′-end. The chart of full-length product elongation efficiency (%) for four PABA types. (**Right**): Gel electrophoresis images of elongation product using Taq DNA polymerase for S-PABA in FAM channel (**a**) and StainAll staining (**b**). Primer/template mixture controls: lane 1 Z0/T, lane 3 Z1-S/T, and lane 13 Z1,2-S/T. Mixtures after elongation: lane 2 Z0/T, lane 4 Z1-S/T, lane 5 Z2-S/T, lane 6 Z3-S/T, lane 7 Z4-S/T, lane 8 Z5-S/T, lane 9 Z6-S/T, lane 10 Z7-S/T, lane 11 Z1,2-S/T, and lane 12 Z1,3-S/T.

**Figure 3 ijms-25-00617-f003:**
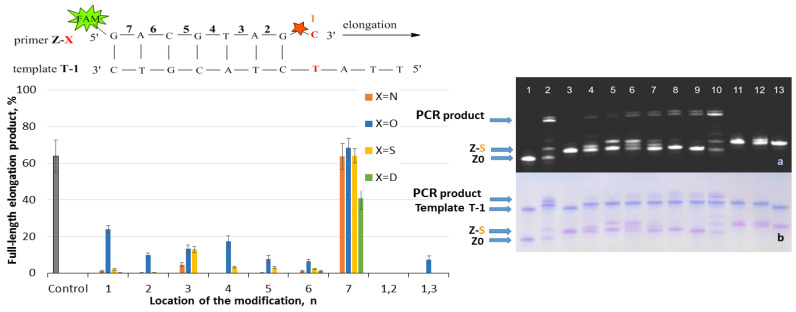
Effect of PABA modification location in a primer on elongation efficiency. (**Left**): Schematic representation of the Z-X/T-1 fragment with a red star of PABA modification with number (n) of phosphates from 3′-end. The chart of full-length product elongation efficiency (%) for four PABA types. For the control mixture, primer Z0 without modification was used. (**Right**): Gel electrophoresis images of elongation product using Taq polymerase for S-PABA in FAM channel (**a**) and StainAll staining (**b**). Primer/template mixture controls: lane 1 Z0/T-1, lane 3 Z1-S/T-1, and lane 13 Z1,2-S/T-1. Mixtures after elongation: lane 2 Z0/T-1, lane 4 Z1-S/T-1, lane 5 Z2-S/T-1, lane 6 Z3-S/T-1, lane 7 Z4-S/T-1, lane 8 Z5-S/T-1, lane 9 Z6-S/T-1, lane 10 Z7-S/T-1, lane 11 Z1,2-S/T-1, and lane 12 Z1,3-S/T-1.

**Figure 4 ijms-25-00617-f004:**
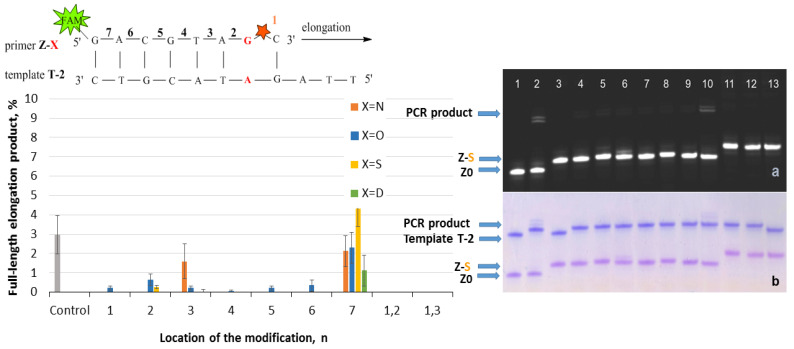
Effect of PABA modification location in a primer on elongation efficiency. (**Left**): Schematic representation of the Z-X/T-2 fragment with a red star of PABA modification with number (n) of phosphates from 3′-end. The chart of full-length product elongation efficiency (%) for four PABA types. For the control mixture, primer Z0 without modification was used. (**Right**): Gel electrophoresis images of elongation product using Taq polymerase for S-PABA in FAM channel (**a**) and StainAll staining (**b**). Primer/template mixture controls: lane 1 Z0/T-2, lane 3 Z1-S/T-2, and lane 13 Z1,2-S/T-2. Mixtures after elongation: lane 2 Z0/T-2, lane 4 Z1-S/T-2, lane 5 Z2-S/T-2, lane 6 Z3-S/T-2, lane 7 Z4-S/T-2, lane 8 Z5-S/T-2, lane 9 Z6-S/T-2, lane 10 Z7-S/T-2, lane 11 Z1,2-S/T-2, and lane 12 Z1,3-S/T-2.

**Figure 5 ijms-25-00617-f005:**
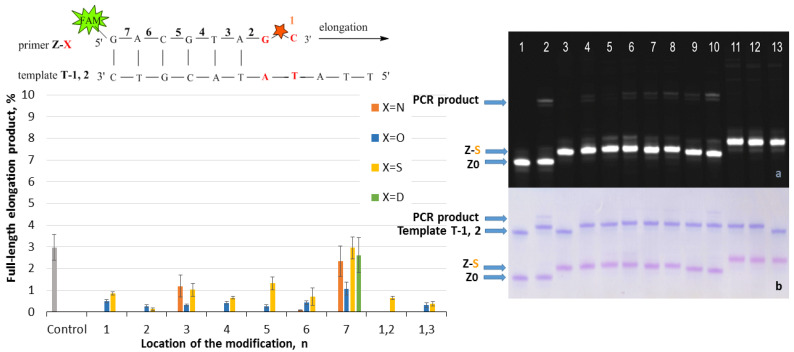
Effect of PABA modification location in a primer on elongation efficiency. (**Left**): Schematic representation of the Z-X/T-1,2 fragment with a red star of PABA modification with number (n) of phosphates from 3′-end. The chart of full-length product elongation efficiency (%) for four PABA types. For the control mixture, primer Z0 without modification was used. (**Right**): Gel electrophoresis images of elongation product using Taq polymerase for S-PABA in FAM channel (**a**) and StainAll staining (**b**). Primer/template mixture controls: lane 1 Z0/T-1,2, lane 3 Z1-S/T-1,2, and lane 13 Z1,2-S/T-1,2. Mixtures after elongation: lane 2 Z0/T-1,2, lane 4 Z1-S/T-1,2, lane 5 Z2-S/T-1,2, lane 6 Z3-S/T-1,2, lane 7 Z4-S/T-1,2, lane 8 Z5-S/T-1,2, lane 9 Z6-S/T-1,2, lane 10 Z7-S/T-1,2, lane 11 Z1,2-S/T-1,2, and lane 12 Z1,3-S/T-1,2.

**Figure 6 ijms-25-00617-f006:**
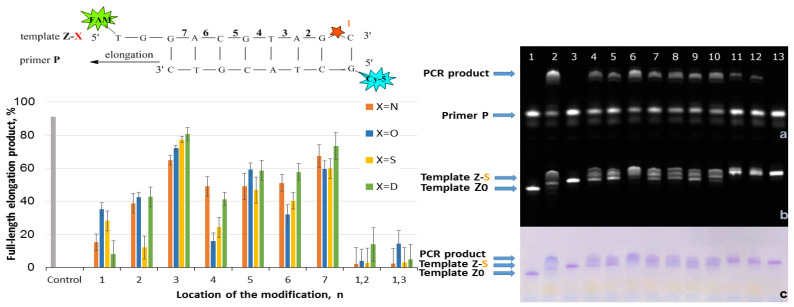
Effect of PABA modification location in a template on elongation efficiency. (**Left**): Schematic representation of the Z-X/P fragment with a red star of PABA modification with a number (n) of phosphates from the 3′-end of the primer. The chart of PCR full-length product elongation efficiency (%) for four PABA types. For the control PCR mixture, template Z0 without modification was used. (**Right**): Gel electrophoresis images of PCR product for S-PABA in Cy5 channel (**a**) FAM channel (**b**) StainAll staining (**c**) Primer/template mixture controls: lane 1 Z0/P, lane 3 Z1-S/P2, and lane 13 Z1,2-S/P. Mixtures after PCR: lane 2 Z0/P, lane 4 Z1-S/P, lane 5 Z2-S/P, lane 6 Z3-S/P, lane 7 Z4-S/P, lane 8 Z5-S/P, lane 9 Z6-S/P, lane 10 Z7-S/P, lane 11 Z1,2-S/P, and lane 12 Z1,3-S/P.

**Figure 7 ijms-25-00617-f007:**
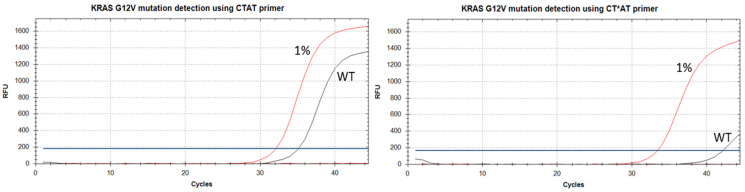
Typical amplification curves obtained in AS-PCR analysis for *KRAS* G12A mutation detection using CT**AT** and CT^S^**AT** primers and 2 × 10^4^ copies of total DNA per reaction for WT and 1% mutant DNA in the background of WT DNA samples. Herein, * means S-PABA modification.

**Table 1 ijms-25-00617-t001:** PCR primer and template sequences.

Abbreviation ^1^	Oligonucleotide Sequence 5′ → 3′
Z 0	FAM-GGTGCGCTCCTGGACGTAGC
Z 1-X	FAM-GGTGCGCTCCTGGACGTAG*C
Z 2-X	FAM-GGTGCGCTCCTGGACGTA*GC
Z 3-X	FAM-GGTGCGCTCCTGGACGT*AGC
Z 4-X	FAM-GGTGCGCTCCTGGACG*TAGC
Z 5-X	FAM-GGTGCGCTCCTGGAC*GTAGC
Z 6-X	FAM-GGTGCGCTCCTGGA*CGTAGC
Z 7-X	FAM-GGTGCGCTCCTGG*ACGTAGC
Z 1,2-X	FAM-GGTGCGCTCCTGGACGTA*G*C
Z 1,3-X	FAM-GGTGCGCTCCTGGACGT*AG*C
T	CTGTTGTTTAGCTACGTCCAGGAGCGCACC
T-1	CTGTTGTTTA**T**CTACGTCCAGGAGCGCACC
T-2	CTGTTGTTTAG**A**TACGTCCAGGAGCGCACC
T-1,2	CTGTTGTTTA**TA**TACGTCCAGGAGCGCACC
P	Cy5-GCTACGTC

^1^ Symbol “*” means phosphoryl N-benzoazole modification location. X—the type of modification (O, S, N, and D). T—template, P—primer, Z—template for Z/P system, and primer for Z/T system. Boldly marked nucleotides in T mean mismatched nucleotides in relation to Z oligonucleotide. PCR conditions are presented in Section 3.2.

**Table 2 ijms-25-00617-t002:** AS-PCR *KRAS* mutation detection using WT DNA (total 2 × 10^4^ copies per reaction) and 1% mutant DNA on the background of WT DNA.

Mutation	Primers	Cq		ΔCq	Primers	Cq		ΔCq
		WT	1%	Cq_WT_ − Cq_1%_ ^1^		WT	1%	Cq_WT_−Cq_1%_ ^1^
**G12A**	**CTTC**	38.0 ± 0.3	32.4 ± 0.1	5.3	CTG**C**	32.8 ± 0.2	31.3 ± 0.2	1.5
	C^O^T**TC**	N/A	32.3 ± 0.1	12.7	C^O^TG**C**	34.9 ± 0.1	32.1 ± 0.2	2.8
	^O^CT**TC**	N/A	33.6 ± 0.1	11.4	^O^CTG**C**	36.4 ± 0.2	32.3 ± 0.2	4.1
	CT^O^**TC**	N/A	32.4 ± 0.1	12.6	CT^O^G**C**	36.8 ± 0.2	32.2 ± 0.2	4.6
	C^S^T**TC**	N/A	32.4 ± 0.2	12.6	C^S^TG**C**	34.0 ± 0.3	31.5 ± 0.2	2.5
	^S^CT**TC**	N/A	33.8 ± 0.2	11.2	^S^CTG**C**	37.0 ± 0.2	31.9 ± 0.2	5.1
	CT^S^**TC**	N/A	32.5 ± 0.1	12.5	CT^S^G**C**	37.1 ± 0.2	31.7 ± 0.2	5.4
	C^N^T**TC**	N/A	33.5 ± 0.4	11.5	C^N^TG**C**	38.2 ± 0.3	32.2 ± 0.2	6.0
	^N^CT**TC**	N/A	36.8 ± 0.5	8.2	^N^CTG**C**	40.3 ± 0.4	32.5 ± 0.2	7.8
	CT^N^**TC**	N/A	38.2 ± 0.2	6.8	CT^N^G**C**	39.3 ± 0.4	33.7 ± 0.2	5.6
	C^D^T**TC**	N/A	33.9 ± 0.1	11.1	CT^D^**TC**	N/A	37.1 ± 0.3	7.9
	^D^CT**TC**	N/A	38.0 ± 0.3	7.1				
**G12V**	CT**AT**	35.4 ± 0.3	32.3 ± 0.1	3.1	CTG**T**	27.4 ± 0.1	27.0 ± 0.1	0.4
	C^O^T**AT**	39.7 ± 0.3	33.1 ± 0.1	6.6	C^O^TG**T**	28.2 ± 0.2	27.9 ± 0.1	0.3
	^O^CT**AT**	29.8 ± 0.2	29.4 ± 0.1	0.4	^O^CTG**T**	29.8 ± 0.3	29.5 ± 0.1	0.3
	CT^O^**AT**	42.4 ± 0.7	34.4 ± 0.1	8.0	CT^O^G**T**	36.5 ± 0.2	33.6 ± 0.1	2.9
	C^S^T**AT**	38.5 ± 0.5	33.2 ± 0.1	5.3	C^S^TG**T**	28.1 ± 0.1	28.1 ± 0.1	0.1
	^S^CT**AT**	39.6 ± 0.2	34.4 ± 0.1	5.2	^S^CTG**T**	32.8 ± 0.3	30.0 ± 0.1	2.8
	CT^S^**AT**	41.1 ± 0.8	33.7 ± 0.2	7.4	CT^S^G**T**	30.2 ± 0.1	30.0 ± 0.1	0.3
	C^N^T**AT**	42.0 ± 0.5	35.4 ± 0.8	6.6	C^N^TG**T**	29.2 ± 0.1	29.1 ± 0.1	0.1
	^N^CT**AT**	42.0 ± 0.8	37.1 ± 0.8	4.9	^N^CTG**T**	33.3 ± 0.1	31.7 ± 0.2	1.6
	CT^N^**AT**	44.3 ± 0.8	37.1 ± 0.8	7.2	CT^N^G**T**	30.5 ± 0.1	30.1 ± 0.1	0.4
	C^D^T**AT**	N/A	37.8 ± 0.6	7.2	^D^CT**AT**	N/A	38.8 ± 0.2	6.2
	CT^D^**AT**	N/A	37.6 ± 0.5	7.4				

No template control (NTC) was undetermined in all the reactions; N/A indicates that no Cq was retrieved for a typical 45-cycle reaction. The symbol “O, S, N, and D” indicates the PABA modification location. Boldly marked nucleotides represent mismatched nucleotides concerning the WT DNA. ^1^ If Cq(WT) = N/A, to calculate ∆Cq = Cq(WT) − Cq(1%) value Cq(WT) = 45 has been used.

**Table 3 ijms-25-00617-t003:** PCR efficiency for four types of PABA modifications, %.

**Primers**	**k-Ref**	**C*TTC**	***CTTC**	**CT*TC**	**C*TGC**	***CTGC**	**CT*GC**
**native**	**100.0**	**92.7**			**93.1**		
O	-	99.3	95.0	97.2	96.8	97.2	95.1
S	-	96.8	96.1	96.5	100.1	96.1	100.1
N	-	99.7	92.4	74.9	94.9	96.7	83.9
D	-	90.0	83.6	80.0	-	-	-
**Primers**	**k-Ref**	**C*TAT**	***CTAT**	**CT*AT**	**C*TGT**	***CTGT**	**CT*GT**
**native**	**100.0**	**93.1**			**92.8**		
O	-	92.1	92.4	97.2	92.3	95.7	96.5
S	-	94.9	97.6	98.8	99.7	98.4	94.9
N	-	84.5	89.0	92.4	97.6	98.4	77.1
D	-	80.5	90.3	93.1	-	-	-

The symbol “*” indicates the PABA modification location. Boldly marked nucleotides represent mismatched nucleotides concerning the WT DNA sequence. PCR efficiency experiments were evaluated using various amounts of DNA. PCR efficiency for each of these primers was calculated using a Thermo Fisher Scientific online calculator (https://www.thermofisher.com/ru/ru/home/brands/thermo-scientific/molecular-biology/molecular-biology-learning-center/molecular-biology-resource-library/thermo-scientific-web-tools/qpcr-efficiency-calculator.html) (accessed on 15 November 2023).

**Table 4 ijms-25-00617-t004:** Primer selection for AS-PCR analysis using four types of PABA modification.

Mutation/ Primers		1% DNA Cq Increase Compared to Native Primer, Cycles				PCR Efficiency, %				ΔCq = Cq_WT_ − Cq_1%_				Conclusion of Suitability for Further AS-PCR Studies			
		O	S	N	D	O	S	N	D	O	S	N	D	O	S	N	D
**G12A**	C*T**TC**	−0.1	0.0	1.1	1.5	99	97	100	90	12.7	12.6	11.5	11.1	++	++	++	+
	*CT**TC**	1.2	1.4	4.4	5.6	95	96	92	84	11.4	11.2	8.2	7.9	++	++	-	-
	CT***TC**	0.0	0.1	5.8	4.7	97	97	75	80	12.6	12.5	6.8	7.1	++	++	-	-
	C*TG**C**	0.8	0.2	0.9	-	97	100	95	-	2.8	2.5	6.0	-	-	-	+	-
	*CTG**C**	1.0	0.6	1.2	-	97	96	97	-	4.1	5.1	7.8	-	+	+	+	-
	CT*G**C**	0.9	0.4	2.4	-	95	100	84	-	4.6	5.4	5.6	-	+	+	-	-
**G12V**	C*T**AT**	0.8	0.9	3.1	5.5	92	95	85	81	6.6	5.3	6.6	7.2	+	+	-	-
	*CT**AT**	−2.9	2.1	4.8	6.5	92	98	89	90	0.4	5.2	4.9	6.2	-	+	-	-
	CT***AT**	2.1	1.4	4.8	5.3	97	99	92	93	8.0	7.4	7.2	7.4	+	+	-	-
	C*TG**T**	−4.4	−4.2	−3.2	-	92	100	98	-	0.3	0.1	0.1	-	-	-	-	-
	*CTG**T**	−2.8	−2.3	−0.6	-	96	98	98	-	0.3	2.8	1.6	-	-	-	-	-
	CT*G**T**	1.3	−2.3	−2.2	-	97	95	77	-	2.9	0.3	0.4	-	-	-	-	-

Red, yellow, and green colors mean “not suitable values”, “average”, and “optimal or excellent”, respectively. The symbol “*” indicates the PABA modification location. Boldly marked nucleotides represent mismatched nucleotides concerning the WT DNA sequence. ++ excellent, + good, - worse.

## Data Availability

Data are contained within the article and Appendix A.

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
