# Peer review of "Phosphoramidate Azole Oligonucleotides for Single Nucleotide Polymorphism Detection by PCR"

_ijms, 2024, doi:10.3390/ijms25010617_

Round 1

Reviewer 1 Report

Comments and Suggestions for Authors

The manuscript by Chubarov et al explores various PABA-related oligonucleotide modifications and locations within a primer and/or template DNA and how they effect PCR processes and target specificity. Overall the manuscript is very thorough and there are only a few concerns to be addressed.

Concerns:

Overall the reading of the work is a bit dense. Reader comprehension of the general material would be enhanced by better presentation of the questions each set of data are answering, including the variations in PCR process, PABA modification, location, etc. Moreover, more specific conclusions throughout the results and discussion section would enhance reader comprehension.

Some things are not clear - such as the Tm choices for the 1x and the 3x cycles of PCR - why extension at 60 or 37? annealing at 30? Explanations would benefit the manuscript.

The utility of the section on template modification is unclear. In general the study is set up to examine SNP prevalance, and modification of native DNA isn't physiologically relevant, so this section (top of page 8) should either be better tied in, or removed.

Lastly, a comparison of the "best" combination (e.g. C*TTC for G12A) of parameters for detection should be compared with the gold standard for current techniques - either in the discussion of the benefits of the author's work, or experimentally. 

Minor concerns:

There were grammatical errors (such as page 2, line 57 "choise" for choice), but they were minor.

Supplemental figure 8's panels have an a-c, but only a and b are described, and clearly the "b" described is "c" - please correct.

Comments on the Quality of English Language

Overall the quality of the language is strong, with few grammatical errors.

Author Response

Thank you for the valuable suggestions and comments. We have carefully examined the comments and suggestions and revised the manuscript accordingly. We presented the word file with track changes. Please find as follows the responses to the comments. Please note that all the comments are bold-faced, and the authors' reply follows immediately below the comments.

Overall the reading of the work is a bit dense. Reader comprehension of the general material would be enhanced by better presentation of the questions each set of data are answering, including the variations in PCR process, PABA modification, location, etc. Moreover, more specific conclusions throughout the results and discussion section would enhance reader comprehension.

Thank you for your suggestion. We have tried to revise the text and makes some additional conclusion through the sections.

Some things are not clear - such as the Tm choices for the 1x and the 3x cycles of PCR - why extension at 60 or 37? annealing at 30? Explanations would benefit the manuscript.

Thank you for question. For 30/20 system, one cycle is enough to see the effect of PABA modification location in a primer on elongation efficiency. For the system 20/8, one cycle is not enough for good reaction elongation to determine any differences between oligonucleotides. For one cycle, for all studied 20/8, the elongation efficiency was low. We increase number of cycles for the possibility of quantitative analysis of elongation results analysis. Such temperatures were used according to the duplex stability.

The utility of the section on template modification is unclear. In general the study is set up to examine SNP prevalance, and modification of native DNA isn't physiologically relevant, so this section (top of page 8) should either be better tied in, or removed.

Thank you for the question. We have tried to revise the text in Section 2.2. follow the reviewer’s question. In the present paper, we have two part of the work: 1) studies about the influence of PABA modification for various application and 2) one important example of application on biomedical system as SNP detection. For the first part, as was mentioned, modification in template takes place in some modern PCR systems.1,2 Moreover, modification will be in one of the chains of DNA in SNP detection system after modified primer elongation (see Picture). Therefore, it is essential to know the effect of modification in template on elongation process, which allow us to study the mechanism.

1         S. Cai, C. Jung, S. Bhadra and A. D. Ellington, Phosphorothioated Primers Lead to Loop-Mediated Isothermal Amplification at Low Temperatures, Anal. Chem., 2018, 90, 8290–8294.

2         P. Srivastava and D. Prasad, Isothermal nucleic acid amplification and its uses in modern diagnostic technologies, 3 Biotech, 2023, 13, 1–23.

 Lastly, a comparison of the "best" combination (e.g. C*TTC for G12A) of parameters for detection should be compared with the gold standard for current techniques - either in the discussion of the benefits of the author's work, or experimentally. 

Thank you for your suggestion. We have tried to revise the text and makes some comparison with literature data.

Minor concerns:

There were grammatical errors (such as page 2, line 57 "choise" for choice), but they were minor.

Supplemental figure 8's panels have an a-c, but only a and b are described, and clearly the "b" described is "c" - please correct.

Thank you for your comment. We have revised the text.

Reviewer 2 Report

Comments and Suggestions for Authors

1. The authors should discuss about the importance of KRAS mutation and the necessity regarding its detection by PCR.

2. Please discuss more about why Phosphoramidate Azole Oligonucleotides was used for showing detection of KRAS mutation.

3. Please explain why the four PABA mutations were chosen?

4. The authors should include a conclusion detailing the utility and future perspective of their work.

Author Response

Thank you for the valuable suggestions and comments. We have carefully examined the comments and suggestions and revised the manuscript accordingly. We presented the word file with track changes. Please find as follows the responses to the comments. Please note that all the comments are bold-faced, and the authors' reply follows immediately below the comments.

  1. The authors should discuss about the importance of KRAS mutation and the necessity regarding its detection by PCR.

Thank you for your suggestion. We have revised the text in the Introduction.

  1. Please discuss more about why Phosphoramidate Azole Oligonucleotides was used for showing detection of KRAS mutation.

Thank you for your suggestion. We have revised the text in the Introduction.

  1. Please explain why the four PABA mutations were chosen?

Thank you for your suggestion. We have revised the text in the Introduction.

  1. The authors should include a conclusion detailing the utility and future perspective of their work.

Thank you for your suggestion. We have revised the text at the end of Results and Discussion section.

Reviewer 3 Report

Comments and Suggestions for Authors

The manuscript describes the use of oligonucleotides carrying uncharged benzoazole phosphoramidate linkages for allele-specific real time PCR (AS-PCR) in the detection of point mutations in K-RAS gene. The authors compared several primers carrying benzoazole phosphoramidate linkages at different positions to increase sensitivity and specificity in the detection of various single nucleotide mutations (SNPs). Results are interesting and the manuscript is well written. 

The synthesis of the modified oligonucleotides was described in a previous paper (ref 28). Although the present manuscript is centered in the use of the modified oligonucleotides in PCR for detection of SNP, I will recommend to add a few words on the synthetic protocol because the modifications used are relatively new and most readers will not know how these oligonclotides can be made. May be authors may include a short phrase describing the introduction of the modification by oxidation with the appropriate azide in the materials and methods under subheading 3.1 synthesis and isolation of oligonucleotides. 

Author Response

Thank you for the valuable suggestions and comments. We have carefully examined the comments and suggestions and revised the manuscript accordingly. We presented the word file with track changes. Please find as follows the responses to the comments. Please note that all the comments are bold-faced, and the authors' reply follows immediately below the comments.

The synthesis of the modified oligonucleotides was described in a previous paper (ref 28). Although the present manuscript is centered in the use of the modified oligonucleotides in PCR for detection of SNP, I will recommend to add a few words on the synthetic protocol because the modifications used are relatively new and most readers will not know how these oligonclotides can be made. May be authors may include a short phrase describing the introduction of the modification by oxidation with the appropriate azide in the materials and methods under subheading 3.1 synthesis and isolation of oligonucleotides. 

Thank you for your suggestion. We have modified Section 3.1 (Synthesis and Isolation of Oligonucleotides).